# Hyperbaric oxygen in children with cerebral palsy: A systematic review of effectiveness and safety

**Justine Laureau[1], Christelle Pons[2,3,4], Guy Letellier[5], Raphaël Gross[1]** *

**1** Nantes Université, CHU Nantes, Movement - Interactions - Performance, MIP, EA 4334, Nantes, France, **2** Pediatric Rehabilitation Department, Fondation ILDYS, Brest, France, **3** Laboratory of Medical Information Processing, LaTIM- INSERM UMR 1101, Brest, France, **4** PMR Department, University Hospital Brest, Brest, France, **5** Paediatric Rehabilitation Center ESEAN-APF, Nantes, France

* raphael.gross@chu-nantes.fr

## Abstract

### Purpose

To report current evidence regarding the effectiveness of hyperbaric oxygen therapy (HBOT) on the impairments presented by children with cerebral palsy (CP), and its safety.

### Materials and methods

PUBMED, The Cochrane Library, Google Scholar, and the Undersea and Hyperbaric Medical Society database were searched by two reviewers. Methodological quality was graded independently by 2 reviewers using the Physiotherapy Evidence Database assessment scale for randomized controlled trials (RCTs) and the modified Downs and Black (m-DB) evaluation tool for non RCTs. A meta-analysis was performed where applicable for RCTs.

### Results

Five RCTs were identified. Four had a high level of evidence. Seven other studies were observational studies of low quality. All RCTs used 100% $O_2$, 1.5 to 1.75 ATA, as the treatment intervention. Pressurized air was the control intervention in 3 RCTs, and physical therapy in 2. In all but one RCTs, similar improvements were observed regarding motor and/or cognitive functions, in the HBOT and control groups. Adverse events were mostly of mild severity, the most common being middle ear barotrauma (up to 50% of children).

### Conclusion

There is high-level evidence that HBOT is ineffective in improving motor and cognitive functions, in children with CP. There is moderate-level evidence that HBOT is associated with a higher rate of adverse events than pressurized air in children.

**Data Availability Statement:** All relevant data are within the manuscript and its Supporting information files.

**Funding:** The author(s) received no specific funding for this work.

**Competing interests:** The authors have declared that no competing interests exist.

## Introduction

Cerebral palsy (CP) is classically defined as a group of permanent movement and posture disorders, responsible for activity limitations, caused by non-progressive injuries to an immature and developing brain [1]. In a recent publication however, CP has been reconsidered, taking advantage of scientific and conceptual advances. In association to the clinical description [1], the concept of early brain injury (EBI) and the post-EBI developmental condition must now be integrated in the care strategies and research management of children with an EBI [2]. Cerebral palsy has proven to have complex aetiology with many risk factors (only recently and partly unravelled). The definition of subtypes of CP, the potential interventions to prevent the extent of lesions and impairments, and the impact of early interventions are still work in progress [2, 3].

The causes of CP are various and often linked with hypoxia conditions in brain tissue [2]. Cerebral palsy is often classified according to the severity of motor activity limitations, using for instance the Gross Motor Function Classification System (GMFCS), which consists of 5 levels of increasing disability [4]. Besides motor dysfunction, cognitive and/or behavioural issues, sensory disorders, seizures, bladder dysfunction, and sleep disorders are often present in CP [1, 2].

Cerebral palsy is the leading cause of motor disability in children [1]. To date, validated interventions regarding motor impairments are symptomatic and the standard of care relies on multidisciplinary interventions, including physiotherapy, occupational therapy, speech therapy, splinting, as well as pharmacological—and in specific cases, surgical—interventions, dedicated to the correction or prevention of orthopaedic consequences [5]. Recent trends within the field of interventions for children with CP include regenerative medicine and alternative therapies, which both benefit from rapidly growing attention among caregivers and families [5, 6]. Considered as a complementary or alternative therapy, hyperbaric oxygen therapy (HBOT) is controversial for many years [5, 7, 8]; HBOT consists in the administration of pure oxygen to a subject placed in a steel or polymeric chamber, with supra-atmospheric pressure (1.5 to 3 atmosphere absolute = ATA). HBOT is part of the hyperbaric therapies (HBT) overall, consisting in the administration of pressurized gas with variable oxygen content and pressure. There are validated indications for HBOT to date [9, 10], among which carbon monoxide poisoning, decompression sickness, arterial gas embolism, and necrotizing soft tissue infections due to anaerobic or mixed bacteriologic agents, are well-known. In CP, an effect of HBOT has been supposed, which relies on a potential physiological effect of increased oxygen rate and/or pressure on the brain cells around the site of injury. This effect is based on several hypotheses, mostly originating from in vitro studies, animal models of neurological anoxo-ischaemic damage, and human studies into stroke. A first hypothesis relies on the presence of quiet neuronal cells in a so-called "ischaemic penumbra" zone, a volume of tissue surrounding the zone of infarction, where cells are viable and can potentially be "switched on" by appropriate stimuli, among which high doses of oxygen [11, 12]. HBOT, by increasing dramatically the amount of free oxygen in the blood, could increase the delivery of oxygen to such areas and reveal the activity of quiet neuronal cells [13]. A second hypothesis is the synaptic sprouting and reorganisation stimulated by the administration of hyperbaric oxygen [13, 14]. A third hypothesis is the presence of stem cells within the brain, which would differentiate into functional neurons in the zones of cerebral damage, a process that could be accelerated by the administration of HBOT after an ischaemic insult to the brain [15, 16]. Other mechanisms, relying on the ischaemia-induced events, such as angiogenesis and enzymatic cascades, have also been proposed to suggest potential effects of HBOT on pathological conditions at a chronic stage, and therefore on CP [17]. Hypoxia would enhance angiogenesis, which would be potentiated by the supply of additional oxygen [18]. Besides, a chemical enzymatic cascade would be triggered by cellular

hypoxia, with a key-role of hypoxia-inducible factor-1a in inflammatory sites with low oxygen levels [19]. Such inflammatory processes induced by anoxia / ischaemia would be down-regulated by HBOT [17]. In contrast with these experimental findings in vitro, there is a lack of studies demonstrating the reversibility of the damages induced by anoxia or ischaemia through HBOT in animal models of CP. In children with CP, the evidence regarding a therapeutic effect of HBOT on motor and/or cognitive impairments remains scarce. To the best of our knowledge, only one literature review has been published (2007) about its effectiveness on motor and cognitive functions in this population [8]. The evidence was considered as inadequate for establishing a significant benefit of HBOT on functional outcomes [8]. The methods and results of the studies that were included in this review have raised important controversies, and calls for more studies, addressing the issue of the control treatment, have been made [20]. Since then, the debate remains open and scientists continue to argue about HBOT in those with CP, mostly through editorials and position papers [5–7, 20]. Recently, Novak et al. endorsed a firm position against the use of HBOT in children with CP. As several clinical studies have been published since the first review [8], it appears relevant to provide an update of the literature regarding the effects of HBOT in children with CP, in order to collect and summarize the evidence on this intervention in this population.

The primary objective of the present systematic review was to assess the effectiveness of HBOT on the impairments presented by children with CP. The secondary objective was to assess the safety of these interventions in children overall.

## Methods

### Search strategy

A systematic review was carried out according to the PRISMA recommendations [21]. The PRISMA checklist assessment for the present systematic review can be found as S1 Checklist. The following databases were searched independently by two authors (JL and RG): PUBMED (from 1949 to May 2021), The Cochrane Library (from 1996 to May 2021), Google Scholar (from 1898 to May 2021), and the Undersea and Hyperbaric Medical Society (UHMS from 1976 to May 2021). The PICO (Patients, Interventions, Comparison, and Outcomes) framework was used to develop literature-searching strategies. Four groups of key-words were designed. The first group was related to the pathology: *cerebral palsy*, *diplegia*, *hemiplegia*, *quadriplegia*. The second group was related to the population: *child*, *children*. The third group was related to interventions: *hyperbaric oxygen therapy*, *hyperbaric oxygenation*, *hyperbaric air*. The fourth group was dedicated to safety and adverse events: *safety*, *adverse events*, *side effects*. A first step was the combination of keywords from groups 1, 2, and 3 for the assessment of effectiveness. A second step was the combination of groups 2, 3 and 4 for the assessment of safety: in order to retain all adverse events potentially occurring in children with CP, all articles dealing with HBOT in children (and not only in children with CP) were retained for this second step. The references of the articles found were also checked to ensure exhaustiveness.

### Article selection: Identification and selection of studies

Two authors (JL and RG) selected the articles independently, based on the titles and abstracts of the references found. The following inclusion criteria were used for study selection:

1. Design: observational studies, before/after studies (descriptive) and controlled trials;

2. Patients: children (<18 years) with CP included in the study sample;

3. Intervention: administration of HBOT or HBA;

4. Outcomes: inclusion of an effectiveness- and/or safety-related outcome for HBOT or HBA;

5. Language: articles written in English.

The exclusion criteria for the articles were: 1) detailed data could not be obtained (only abstracts available), 2) animal model studies; 3) single case-reports, opinion articles, editorials, letters to the editors/comments, and review-articles.

**Quality assessment and study appraisal: Assessment of the characteristics of included studies.** *Quality*. The methodological quality of each article was independently evaluated by 2 authors (JL and RG). Any disagreement resulting from this first assessment was resolved by consensus. The quality of randomised controlled trials (RCTs) was evaluated using the Physiotherapy Evidence Database Assessment Scale (PEDro) [22]. The quality of a study was classified as 'high' ($\geq$6/10), 'fair' (4–5/10) or 'poor' ($\leq$3/10) [23]. Non-RCTs were evaluated using the modified Downs and Black (m-DB) assessment tool [24], which has been reported to be suitable for use in systematic reviews of effectiveness. This tool consists of 27 items which measure internal and external validity, bias and power. The maximum score is 28. For each item of the aforementioned scales, points were only attributed if the criterion could be explicitly found in the text of the article.

The bodies of evidence were then appraised using the GRADE evidence rating system [25]. The GRADE system rates the quality of the evidence (from high to very low) and the strength of recommendation for clinical application. *Participants*: information about participants (age, type of cerebral palsy, and functional level) were detailed.

*Intervention*. data regarding the hyperbaric intervention (content: pure oxygen or air, pressure, number and duration of sessions, total volume of the intervention) and control intervention were carefully searched through and reported.

*Outcome measures*. all types of outcome measures identified in the retained articles were analysed and could subsequently be sorted as: motor function, spasticity, sleep disorders, cognitive functions, independence, for effectiveness-related outcome, and safety.

## Data synthesis

Data regarding the authors, year of publication, population, primary outcome measure, main results, and quality, and data related to interventions (dose, number and duration of hyperbaric treatments, total duration of treatment, and associated interventions), as well as the outcomes, were extracted by two authors (JL and RG) from each article and are presented in Table 1.

Data related to safety and adverse events are shown in Table 2. Adverse events were classified into 4 categories (mild, moderate, severe, and sentinel) according to the usual scale [26].

A meta-analysis was performed when allowed by the treatment protocols and outcome measures (i.e. quantitative outcomes using the same measurement tool, in two or more intervention groups, in at least two different studies). The results of the meta-analysis were expressed using the mean difference and its 95% confidence interval for effectiveness and using odds-ratio and its 95% CI for safety (proportion of adverse events).

The review has been registered in the international prospective register of systematic reviews (PROSPERO, REF 193424).

## Results

### Selection process

The initial automatic database search yielded 6114 articles (Pubmed, N = 30; The Cochrane Library, N = 17, Google Scholar, N = 51, The UHMS, N = 6011 articles, other sources/article

**Table 1. Synthesis of the 12 studies selected for effectiveness: Methodological data, authors, year of publication, population, primary outcome measure, main results, and quality, data related to interventions (dose, number and duration of hyperbaric treatments, total duration of treatment, and associated interventions).** NA: not available. m-DB: modified Downs & Black scale.

| Authors | Design | HBOT dosage | Therapy protocol (number of treatments, duration of each treatment, protocol type of hyperbaric chamber) | Control intervention (and type) | Adjunct therapy | Population | GMFCS | Outcome measure (s) | Results | PEDro score | m-D&B score |
|---|---|---|---|---|---|---|---|---|---|---|---|
| Machado, 1999, Brasil [38] | Retrospective | 100% O$_2$ 1.5 ATA | 20 treatments 60–120 min Protocol NA Monoplace chamber | No | - | 230 children Cerebral palsy Mean age: 7,5 years | NA | Spasticity (no scale described) | Reduction of spasticity in 95% of patients | - | 3 |
| Montgomery et al., 1999, Canada [32] | Observational (before/after) | 95% O$_2$, 1.75 ATA | 20 treatments 60 min 1 treatment / day, 5 days / week, 4 weeks Multiplace chamber | No | - | 25 children Cerebral Palsy (spastic diplegia) Mean age: 5.7 years | NA | GMFM Jebsen-Taylor Test spasticity (MAS) video analysis before and after intervention parents administered questionnaire | Mean improvement on the GMFM = 5.3%. Improvement in 3 out of 6 dimensions on the Jebsen-Taylor test. reduction in spasticity in some muscle groups video analysis post-test: improved motricity in 67% of patients | - | 10 |
| Collet et al., 2001, Canada [27] | RCT | 100% O$_2$, 1.75 ATA | 40 treatments 60 min 1 treatment / day, 5 days / week, 8 weeks Monoplace or multiplace chamber | 21% O$_2$, 1.3 ATA | - | 111 children Cerebral palsy Mean age: 7.5 years | NA | GMFM Language/ Speaking: Dudley/ Delage test, Bleile +/- Bleile and Miller Orofacial functions: Montréal University Protocol Voice: Kent Protocol Verbal and visiospatial working memory: Corsi blocks, pictures & words span test Visual and auditory attention: TOVA PEDI | Improvements of the GMFM in both groups Improvements of language, memory and attention in both groups Improvements of the PEDI score No between-group differences | 8 | - |

*(Continued)*

Table 1. (Continued)

| Authors | Design | HBOT dosage | Therapy protocol (number of treatments, duration of each treatment, protocol, type of hyperbaric chamber) | Control intervention (and type) | Adjunct therapy | Population | GMFCS | Outcome measure(s) | Results | PEDro score | m-D&B score |
|---|---|---|---|---|---|---|---|---|---|---|---|
| Chavdarov, 2002, Bulgaria [33] | Observational (before/after) | 100% O₂, 1.5 to 1.7 ATA | 20 treatments 40–50 min 1 treatment / day, 7 days / week, 20 days Multiplace chamber | No | Vitamin C | 50 children Cerebral palsy Mean age: 10.4 years | I N = 20 II N = 10 III N = 3 IV N = 10 V N = 7 | Motor Abilites: Holt's Assessment of Motor Abilities Mental Abilities: Munich's Functional Developmental Diagnostic, Wechsler's test Raven's test, Frostig's test, Goppinger's test Speech abilities: Nancie Finnie's questionnaire and Wechsler's test | Motor Abilites: proportion of patients who improved = 41.3% Mental Abilities: proportion of patients who improved = 35.3% Speech Abilities: proportion of patients who improved = 43.9% | - | 7 |
| Hardy et al., 2002, Canada [28] | RCT | 100% O₂, 1.75 ATA | 40 treatments 60 min 1 treatment / day, 5 days / week, 8 weeks Monoplace or multiplace chamber | 21% O₂, 1.3 ATA | - | 111 children (75 for neuropsychological tesing) Cerebral palsy Mean age: 8 years | NA | Visiospatial and verbal working memory: Corsi blocks, pictures & words span tests Visual and auditory attention: TOVA Parents' questionnaire: CPRS-R | Same as [27] Significant reduction of 1 item out of 13 on the CPRS-R, (anxiety//) in the treatment group vs 8/13 in the control group, with no between-group difference | 8 | - |
| Sethi et al., 2003, India [34] | Observational | 100% O₂, 1.75 ATA | 40 treatments 60 min 1 treatment / day 6 days/week Multiplace chamber | No | Occupational therapy | 30 children Cerebral Palsy Mean age: 4 years | NA | Motor Capacity: Norton's Basic Motor Evaluation Scale | Significant improvement in both groups, no statistical analysis of between-group difference. | - | 5 |
| Hegazy et al., 2006, Egypt [30] | RCT | 100% O₂, 1.7 ATA | 75 treatments 60 min 1 treatment / day, 5 days / week, 12 weeks Multiplace chamber | Physical therapy | Physical therapy | 40 children Cerebral palsy Mean age: 5.5 years | NA | Spinal reflexes (H/ M ratio in the soleus) Gross manual dexterity: Box and blocks test Inspiratory capacity (spirometer) | Improvement of the outcomes in both groups; No between-group difference | 6 | - |

(Continued)

**Table 1.** (Continued)

| Authors | Design | HBOT dosage | Therapy protocol (number of treatments, duration of each treatment, protocol, type of hyperbaric chamber) | Control intervention (and type) | Adjunct therapy | Population | GMFCS | Outcome measure(s) | Results | PEDro score | m-D&B score |
|---|---|---|---|---|---|---|---|---|---|---|---|
| Lacey et al., 2012, USA [29] | RCT | 100% O₂ 1.5 ATA | 40 treatment 60 min 1 treatment / day, 5 days / week, 8 weeks Multiplace chamber | 14% O₂, 1.5 ATA | - | 49 children Cerebral palsy Mean age: 5.5 years | I N = 10 II N = 6 III N = 6 IV N = 9 V N = 22 | GMFM-88 PEDI score TOVA | GMFM: No change in any group PEDI: significant improvement in 4 out of 6 domains in the HBOT group, in all domains in the control group, no between-group difference. TOVA: No change in any group | 10 | - |
| Mukherjee et al., 2014, India [35] | Observational | Group B: 21% O₂, 1.3 ATA Group C: 100% O₂ 1.5 ATA Group D: 100% O₂ 1.75 ATA | 40 treatments 60 min 1 treatment / day, 6 days / week, for a total of 40 treatments Multiplace chamber | Group A: Physiotherapy alone | - | 150 children Cerebral palsy Mean age: 4.3 years | NA | GMFM-66 | Improvement of the GMFM-66 score in all groups; Improvement was significantly higher in the HBOT groups than in the HBA group No between-group differences for the various pressure dosages in the HBOT groups | - | 13 |
| Long et al., 2017, China [37] | Observational (before/after) | 100% O₂ 1.6 ATA | 15–20 treatments 60 min 1 treatment / day, 5 days / week, 3 to 4 weeks Multiplace chamber | No | - | 71 children Cerebral palsy Mean age: 4 years | I N = 16 II N = 22 III N = 13 IV N = 4 V N = 16 | Total Sleep Items from the SDSC scale | Improvement (reduction) of the TSI after 10 and 20 HBOT treatments | - | 14 |

(Continued)

Table 1. (Continued)

| Authors | Design | HBOT dosage | Therapy protocol (number of treatments, duration of each treatment, protocol, type of hyperbaric chamber) | Control intervention (and type) | Adjunct therapy | Population | GMFCS | Outcome measure (s) | Results | PEDro score | m-D&B score |
|---|---|---|---|---|---|---|---|---|---|---|---|
| Jovanovic et al., 2017, Serbia [36] | Observational (before/after) | 100% O$_2$, 1.55 ATA | 120 treatments 60 min 1 treatment / day, 5 days / week, for a total of 30 treatments, then a 6-week pause, and so on Device: NA | No | - | 52 children Cerebral palsy Mean age: 5 years | II N = 30 V N = 22 | SDSC scale GMFM-88 Verbal production | SDSC: improvement from 60 treatments in the GMFCS V children, and from 30 treatments in the GMFCS II children GMFM: GMFCS V group showed improvement in the lying with rotation and sitting items, from 30 treatments. GMFCS II group showed improvement in the standing and walking items from 120 sessions | - | 11 |
| Azhar et al., 2017, Pakistan [31] | RCT | 100% O$_2$, 1.5 ATA | 200 treatments 60 min 5 sessions of 40 treatments: 1 treatment / day, 5 days / week, 8 weeks Multiplace chamber | Physical Therapy | - | 200 children Cerebral palsy Mean age: 9.5 years | NA | GMFM score Barthel index Parents satisfaction (PSI) | GMFM improved in both groups, significantly more in the HBOT group. Barthel Index improved in both groups, significantly more in the HBOT group. PSI improved in both groups | 4 | - |

**Table 2. Adverse events recorded in the studies.**

| Study | Study Type | Number of subjects included | Age (years, min-max) | Pathologies included and numbers | Number of HBOT treatments | Number of adverse events recorded | Description of adverse events and proportion of the total sample |
|---|---|---|---|---|---|---|---|
| **Machado 1989** [38] | Retrospective (1985–1989) | 230 | 0–16 | CP | >4600 | 4 | Seizures (1.7%) |
| **Keenan et al. 1998** [44] | Retrospective (1985–1995) | 32 | 0–11 | Necrotizing infection (N = 21) CO poisoning (N = 9) Air embolism (N = 2) All patients were mechanically ventilated | Not Available | 39 | Hypotension (62.5%) Bronchospasm (34.4%) Hypoxemia (6.3%) Hemotympanum (12.5%) Extubation (3.1%) Seizures (3.1%) (8 children underwent prophylactic myringotomy) |
| **Waisman et al. 1998** [45] | Retrospective (1980–1997) | 139 | 0.2–18 | CO poisoning (N = 111) Crush injury, Traumatic Ischaemia or Compartment Syndrome (N = 13) Muscular Necrosis due to Clostridium Infections (N = 4) Refractory osteomyelitis (N = 5) Massive air embolism (N = 2) Purpura Fulminans (N = 2) Decompression Sickness (N = 1) Necrotizing Fasciitis (N = 1) | Not Available | 2 | Seizure: 1 case (0.7%) Pulmonary toxicity of $O_2$: 1 case (0.7%) (Tympanic membrane paracentesis was performed in 22 children before HBOT) |
| **Montgomery et al 1999** [32] | Observational (before/after) | 25 | 3–8 | CP (spastic diplegia) | 20 | 0 | 13 subjects underwent prophylactic tympanostomy tube placement |
| **Nuthall 2000** [57] | Case report | 2 | 4 and 10 | CP | Not Available (case reports) | | Acute respiratory failure followed by seizures |
| **Collet et al. 2001** [27] | RCT | 111 | 3–12 | CP | 4440 | 57 | Ear problems HBOT group: N = 42 (37.8%) HBA group: N = 15 (13.5%) |
| **Chavdarov 2002** [33] | Observational (before/after) | 50 | 1–19 | CP | 1000 | 4 | HBOT was stopped in 4 cases (8%) because of adverse events (seizures, oral movement disorders, facial sensory disorders, tachycardia) |

(*Continued*)

**Table 2.** (Continued)

| Study | Study Type | Number of subjects included | Age (years, min-max) | Pathologies included and numbers | Number of HBOT treatments | Number of adverse events recorded | Description of adverse events and proportion of the total sample |
|---|---|---|---|---|---|---|---|
| **Muller-Bolla et al. 2006** [41] | RCT | 111 | 3–12 | CP | 4440 | HBOT group (N = 57): 95 adverse events recorded. 50% of children had at least a MEBT. Control group (HBA 21% $0_2$ 1.3 ATA, N = 54): 64 AE recorded. 27.8% of children had at least a MEBT. | Middle Ear Barotrauma: HBOT 50% of children, HBA 27.8%<br>Pharyngitis: HBOT 28.6%, HBA 14.8%<br>Ear pain: HBOT 14.3%, HBA 13%<br>Otitis: HBOT 7.1%, HBA 7.4%<br>Fever: HBOT 5.4%, HBA 5.6%<br>Dyspepsia: HBOT 1.8%, HBA 7.4%<br>Others: < 5% in both groups. (3 children underwent prophylactic myringotomy in the HBOT group, 0 in the HBA group) |
| **Lacey et al. 2012** [29] | RCT | 49 | 3–8 | CP | 1960 | 20 | HBOT (24 patients): Ear pain N = 7 (29%)<br>Fluid in ear/nose + rectal bleeding N = 1 (4.2%)<br>HBA group (22 patients): Ear pain N = 8 (36%)<br>Seizure N = 1 (4.5%) not related to HBA. |
| **Frawley et al. 2012** [43] | Retrospective (1998–2010) | 54 | 0–16 | Decompression illness (N = 4)<br>Cerebral arterial gas embolism (N = 2)<br>Acute arterial ischaemia (N = 21)<br>Clostridial infection (N = 3)<br>Necrotizing fasciitis (N = 7)<br>Meningococcal *Purpura fulminans* (N = 3)<br>Nonhealing wounds (N = 5)<br>Radiation soft tissue injury (N = 4)<br>Refractory osteomyelitis (N = 2)<br>Avascular necrosis femoral head (N = 1)<br>Complex regional pain syndrome (N = 1)<br>Oxycephalies (N = 1) | 668 | 44 | Hypotension N = 4 (3 USI,1 NUSI) (7.4%)<br>Progressive hypoxemia N = 2 (2 USI/ 0 NUSI) (3.7%)<br>Neurological Toxicity on the central nervous system N = 2 (1 USI/1 NUSI) (3.7%)<br>Anxiety N = 15 (4 USI, 11 NUSI) (27.7%)<br>Nausea N = 15 (2 USI/ 13 NUSI) (27.7%)<br>MEB N = 6 (6 NUSI) 11.1% (12 children underwent prophylactic myringotomy) |

*(Continued)*

**Table 2.** (Continued)

| Study | Study Type | Number of subjects included | Age (years, min-max) | Pathologies included and numbers | Number of HBOT treatments | Number of adverse events recorded | Description of adverse events and proportion of the total sample |
|---|---|---|---|---|---|---|---|
| **Frawley et al. 2013** [46] | Retrospective (1998–2011) | 112 | 11–16 | Decompression illness (N = 9) Cerebral arterial gas embolism (N = 4) Compartment syndrome (N = 13) Crush injury (N = 14) Iatrogenic acute arterial ischaemia (N = 7) Clostridial infection (N = 4) Necrotising fasciitis (N = 10) Fungal sepsis (N = 3) Purpura fulminans (N = 3) CO poisoning (N = 9) Surgical wounds (N = 6) Mycobacterium ulcerans (N = 2) Crohn's disease (N = 3) Radiation soft tissue injury (N = 8) Refractory osteomyelitis (N = 13) CP (N = 1) Optic neuropathy (N = 1) Chronic regional pain syndrome (N = 1) Oxycephalus (N = 1) | 1099 | 58 | MEB N = 4 (3.6%) Anxiety N = 22 (19.6%) Seizures N = 3 (2.7%) Pulmonary toxicity of $O_2$ N = 3 (2.7%) Nausea N = 18 (16.1%) Progressive hypoxemia N = 3 (2.7%) Hypotension N = 4 (3.6%) Symptomatic hypoglycemia N = 1 (0.9%) |
| **Mukherjee et al. 2014** [35] | Observational (case/control) | 150 | > 10 | CP | | 3 | Three children were excluded because of ear pain on compression. |
| **Hadanny et al. 2016** [42] | Retrospective (2010–2014) | 240 children (10.3% of the total sample = 2334 patients) | 0–16 | various | Total number = 62614 | 452 | MEB N = 215 (9.2%) Ear or sinus pain without objective barotrauma N = 79 (3.4%) Sinus squeeze N = 16 (0.7%) Seizures N = 7 (0.3%) Neurological Toxicity of $O_2$ N = 1 (0.04%) Hypoglycemia N = 9 (0.4%) Dizziness/sickness N = 36 (1.5%) Claustrophobia N = 6 (0.3%) Chest pain N = 22 (0.95%) Dyspnea N = 8 (0.3%) Visual disorders N = 8 (0.3%) |
| **Long et al. 2017** [37] | Observational (before/after) | 71 | 2–6 | CP | 1138 | 14 | Ear pain N = 12 (16.9%) Seizures N = 2 (2.8%) |

*(Continued)*

**Table 2.** (Continued)

| Study | Study Type | Number of subjects included | Age (years, min-max) | Pathologies included and numbers | Number of HBOT treatments | Number of adverse events recorded | Description of adverse events and proportion of the total sample |
|---|---|---|---|---|---|---|---|
| **Azhar et al. 2017** [31] | RCT | 200 | 5–14 | CP and TD | 19400 | Not available | 3.5% of patients (N = 7) experienced minor side effects, described as ear pain and/or confinement anxiety. Numbers in each group (CP and TD) not available |

references, N = 5). Eventually, 12 articles were retained, including 5 RCTs and 7 observational studies, written between 1989 and 2017. Data regarding the study selection process are summarized in the Flow-Chart (Fig 1).

## Quality assessment

Out of the twelve studies, five were RCTs, with four high-quality RCTs [27–30] and one fair-quality RCT (PEDro score = 4 [31]). The other seven studies were observational studies. Six were prospective descriptive studies with a m-DB score between 5 and 14 [32–37], and one was a retrospective study with a m-DB score of 3 [38]. The details of the PEDRO score calculations for the different RCTs can be found in S1 File, and the details of the MDB scores of the non-RCT studies can be found in S2 File.

## Description of the studies

A total of 1008 children were included in these 12 studies, all suffering from CP. The age of the children included in the different studies ranged from 4 to 10.4 years. The HBOT protocol used (95–100% $O_2$, 1.5 to 1.75 ATA) varied from 20 to 200 sessions, lasting for at least 1 hour, 5 to 7 days / week for 3 to 12 weeks, depending on the study. Controlled studies (N = 6) used hyperbaric air (HBA) (N = 4) or physical/occupational therapy (N = 2) as the control intervention. Data regarding the population, protocols, treatment and control interventions are specified in Table 1.

## Outcomes

Various outcomes were assessed in the 12 retrieved studies.

Nine studies reported data about motor function as a primary outcome measure [27, 29–35, 38]. The GMFM was used as the main outcome measure in five out of these nine studies [27, 29, 31, 32, 35].

Spasticity was assessed in three studies [30, 32, 38], cognitive functions in five [27–29, 33, 36], sleep disorders in three [36, 37], and functional performance, using the PEDI scale, in two [27, 29]. See Table 1.

## Effectiveness according to the outcome

**Motor function (9 studies).** Four RCTs reported the effectiveness of HBOT on motor function. All used the GMFM score, considered as the gold-standard for the assessment of motor function in children with cerebral palsy [39]. The main result of two high-quality RCT [27, 29] (PEDRo score = 8 and 10) was a similar improvement of the GMFM score in the

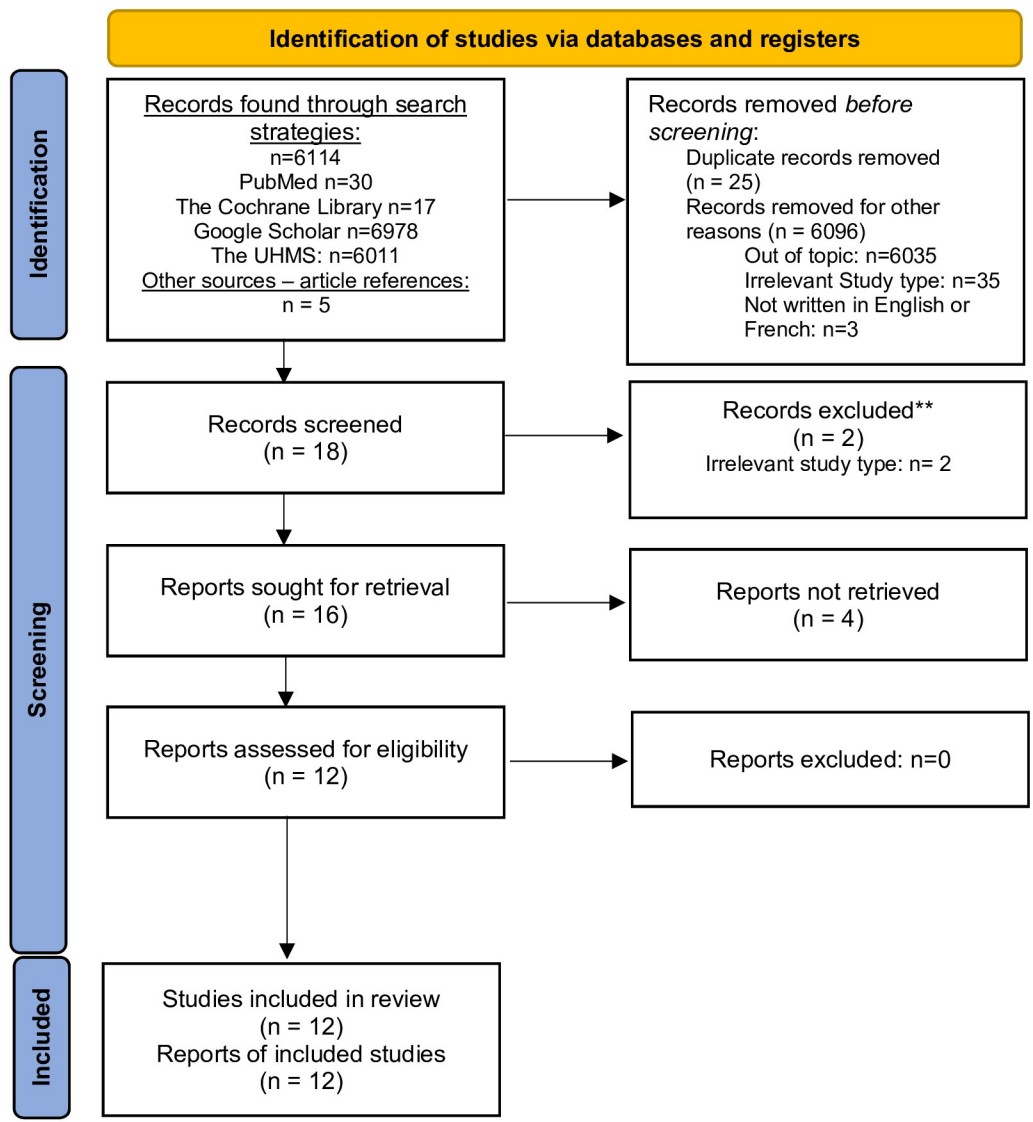

**Fig 1. Flow-chart of the studies retained for the assessment of effectiveness of hyperbaric oxygen therapies in children with CP (primary objective).**

intervention and the control group (pressurized air) (+2.9 points in the HBOT group, + 3.0 points in the control group, p = 0.544) in one RCT [27], and no change in the GMFM score in any group with no between-group difference in another RCT [29]). One RCT (PEDRo score = 6,) investigated upper limb function [30], and found significant improvements of manual dexterity in both groups (assessed by the Box and Blocks test), with no between-group differences.

One recent, fair-quality RCT [31] (PEDro score 4/10,) reported a significant improvement of motor function but without providing the actual GMFM scores (96% of patients improved on the GMFM in the HBOT group vs. 37% in the control group (physiotherapy) p<0.0001).

A meta-analysis was performed based on the results from the two studies describing the pre- and post- intervention GMFM scores for HBOT *vs.* a control intervention [27, 29]. No difference in the GMFM score change was found (mean difference = 0.14 [-1.00;1;29], see Fig 2).

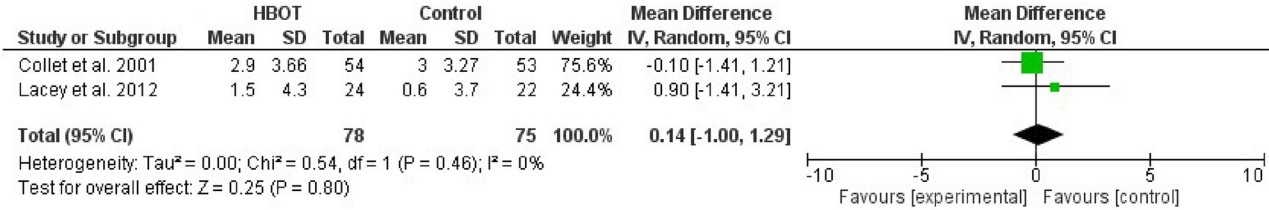

**Fig 2. Meta-analysis: Forest-plot of the differences in outcomes for the GMFM score in the selected studies.**

Besides these four RCTs, two observational studies assessed motor function in children with CP, using the GMFM score as an outcome measure (*See* Table 1). One observational study ([35], mDB = 13) compared HBOT to occupational therapy and reported an improvement of the GMFM and its sub scores, while the study by Montgomery [32] (m-DB = 10) reported no improvement on the GMFM overall but with 3 categories of the GMFM out of 5 being significantly improved.

Two observational studies [33, 34], used motor function as a primary outcome measure, but without using the GMFM score (see Results in Table 1).

**Spasticity (3 studies).** Three studies [30, 32, 38] included spasticity assessments. One RCT [30] reported a significant reduction of spasticity as assessed by the H/M ratio in the *soleus* muscle in both the experimental and control groups, without any between-group difference. Two before/after studies [32, 38] (m-DB 3–10) reported a reduction of spasticity, assessed clinically through the MAS [32] or a subjective scale [38]. *See* Table 1.

**Cognitive functions (5 studies).** Cognitive functions were assessed in three RCTs using various outcome measures [27–29]: Visiospatial and verbal working memory and attention [27, 28], or Test of Variables of Attention [29]. HBOT did not provide additional benefit relative to pressurized air in children with CP for the cognitive and psychological functioning, and overall independence, in any study. *See* Table 1.

Two before/after studies [33, 36] (m-DB = 10 and 11) found a significant improvement in cognitive functions (up to 35% of children based on Mental Abilities and speech abilities as outcome measures in [33]) or language based on the spontaneous verbal production (up to 44% of children [36]).

**Sleep disorders (3 studies).** Three observational studies investigated the impact of HBOT on sleep disorders (m-DB = 10–14), [32, 36, 37]. The main outcome measure was the total sleep items (TSI) from the Sleep Disturbance Scale for Children Questionnaire [40] in 2 out of 3 studies [36, 37]. Both reported a significant reduction of the TSI after HBOT. Montgomery et al. [32] used a custom questionnaire completed by parents, and reported an improvement in the quality of sleep. *See* Table 1.

**Independence (4 studies).** In the two high-quality RCTs [27, 29] investigating the functional performance of children with CP using the PEDI (Pediatric Evaluation of Disability Inventory) score, an improvement was reported in both the experimental group and the control group, without any between-group difference.

**Safety (5 studies).** Six studies assessed the adverse events occurring during / after HBOT in children [41–46] as a primary outcome measure. One of these studies specifically investigated the adverse effects of HBOT in children with CP [41], in a safety analysis based on the RCT by Collet et al. [27]: HBOT 1.75 ATA vs 1.3 ATA hyperbaric air. Middle ear barotrauma (MEB) was significantly more frequent in the HBOT group (50% of patients *vs.* HBT: 27.8%, relative risk = 1.5, 95% IC = 1.1–2.2, p = 0.02). A multivariate analysis did not find any factor associated with a higher risk of barotrauma. Other adverse events were rare and did not differ

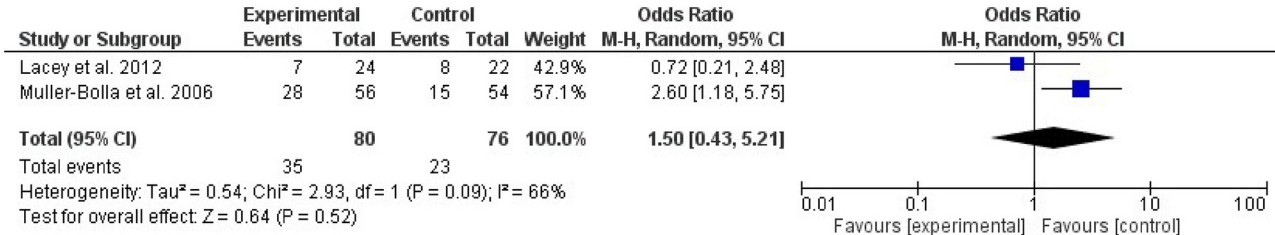

**Fig 3. Meta-analysis of safety: Forest-plot of the differences in adverse events occurrence in the selected studies.**

between groups. Data from the other studies [29, 42–46] indicated that overall, the adverse events were mostly rare and of mild severity (*See* Table 2). The most frequent was middle ear barotrauma (up to 50% of cases) [41]. Less frequently, seizures, confinement anxiety, pulmonary disorders, nausea, hypoglycemia, hypotension, visual disorders and dizziness were observed. Younger age was likely to be a risk-factor for middle-ear barotrauma: in the study by Hadanny et al. [42] about adverse events in a retrospective series of 2334 patients treated with HBOT (240 children), a multivariate analysis showed that subjects under the age of 16 had a higher risk of evidenced middle ear barotrauma (adjusted odd-ratio = 2.729, 95% CI = 1.352–5.505, p = 0.005).

A meta-analysis was performed based on the results from the two controled studies describing the adverse events of HBOT *vs.* a control intervention in cerebral palsy [29, 41]. No increase in the relative risk was found in the HBOT group. (OR: 1.50 [0.43–5.21], see Fig 3).

Other studies assessed effectiveness of HBOT as the primary outcome measure (see above) but also reported data about adverse events. The results from the studies reporting safety are listed in Table 2.

## Discussion

The results from the present systematic review are the following: first, based on three high-quality RCTs [27, 29, 30], there is high level evidence that HBOT is ineffective in improving motor function, cognitive functions, and functional performance, compared to pressurized air or physical therapy. Secondly, based on three observational studies [32, 36, 37] there is very low level evidence that HBOT improves sleep disorders in children with CP. Thirdly, based on two prospective studies assessing the adverse events in a systematic way [29, 42], there is conflicting evidence about whether HBOT is associated with a higher rate of side effects than 1.3 ATA air. Given these results, HBOT is not recommended in clinical practice in children with CP. The present review confirms previous conclusions that hyperbaric interventions should not be administered in standard care [5, 7]. These results are in accordance with the previous systematic review, published in 2007 [8], and with the recent report endorsed by the Québec government [47].

### Interventions and controls

The main result of the present systematic review is the high-level evidence that HBOT does not improve motor function compared to control interventions, though outstanding results have been reported in a lower-quality RCT [31], or in observational studies. It has to be noticed that the control intervention was variable among the 4 RCT assessing motor function: hyperbaric air in two [27, 29] and physical and/or occupational therapy in two [30, 31]. There has been considerable debate about the issue of the control intervention. Authors who claim for efficacy of HBOT in children with CP have stressed that the negative results of HBOT in

the literature have to be reconsidered based on a potential effect of hyperbaric air, which is frequently used as the control intervention due to its similarity in the material conditions and pressure effect [48]. Pressures of about 1.3 ATA, which is the threshold from which the subject in the chamber can perceive mechanical and physiological effects, are classically used as the control, "sham" procedures in studies on HBOT. However, physiological effects have been evidenced at such pressure levels [49]. This point has led some authors to claim that the improvements in the outcomes that were observed in both the HBOT and control groups could be due to a genuine physiological effect of hyperbaric air rather than to a placebo effect [27, 48]. Then, Lacey et al., designed their RCT in order to compare HBOT to hyperbaric air with reduced oxygen rate (1.5 ATA, 14% $O_2$ to obtain physiological blood oxygen content), and obtained similar negative results in both groups [29]. This study has been used by the majority of the paediatric community to consider the debate as ended [5, 7]. However, given that any increase in pressure, even with reduced or adjusted oxygen rates, must be considered as having a potential physiological effect, its use as a placebo remains challenged, together with the conclusions of Lacey et al. [29]. Theoretically, it is possible to design future studies with an experimental design where the effects of hyperpressurization and hyperoxygenation would be differentiated, by dividing the experimental samples into, for instance, four experimental groups: one group with usual HBOT (1.5 ATM, 100% $O_2$), one group with pure oxygen at normal pressure (1.0 ATM, 100% $O_2$), one group with pressurized air and an equivalent atmospheric oxygen content (1.5 ATM, 14% $O_2$), and one control group with normal air (1.0 ATM, 21% $O_2$).

## Patients

All types of clinical presentations of CP were included in the different studies of this systematic review. The influence of the vast heterogeneity of patients on the negative results in the RCTs included in the present review can be questioned. Dispersion is also apparent regarding the severity of the CP. The GMFCS level was available in only 4 out of 12 studies, and was never considered as an inclusion criterion. A wide range of age is also apparent in the studies. Finally, the aetiology of the CP and pathophysiological mechanisms of the brain injury in the children included were hardly ever detailed. Yet, given the physiological effects of hyperbaric treatments, brain lesions where hypoxia at the brain tissue level is the limiting factor of recovery have been considered as the target of HBOT in neurological indications [9]. Asl et al. [50] (not included in the present review because no clinical outcomes were measured) assessed brain perfusion, using SPECT, in 4 patients with CP aged 5 to 27 years, before and after 40 treatments with HBOT. Two patients out of four showed improved cerebral perfusion after HBOT. In stroke patients, Efrati et al. [51] showed that improvements in impairments and activity after HBOT correlated with changes in SPECT imaging, and that elevated brain activity was detected mostly in regions of alive cells (as confirmed by CT) with low activity (based on SPECT). However, the level of evidence of HBOT in other brain injuries than CP (for instance, in adult patients with acquired lesions) remains low [10, 14, 52, 53]. It could be proposed in future studies to assess brain perfusion imaging and functional MRI, together with clinical measures, in order to determine if a sub-type of cerebral lesion (leukomalacia, stroke. . .) could be associated with a positive response to HBOT, which would be in line with its presumed effect of increasing the delivery of oxygen to the brain.

## Developmental trajectories

HBOT is usually delivered on a mid or long-term period (usually 40 treatments distributed on approximately 2 months, but protocols with up to 200 treatments [31] have been used, see Table 1). With time, children with CP present a spontaneous evolution of their motor

performance. In EBI and CP, the developmental trajectories must be considered, as the functional performance level varies with time, even if the brain lesions are stabilized [54]. Regarding motor function for instance, the GMFM-ER (Gross Motor Function Measure Evolution Ratio) has been proposed as an outcome measure to take the natural progression of CP children into account [55]. It is possible that this natural progression participated in the improvements observed in both groups in the RCT by Collet et al. [27] or in observational studies. From now on, we assume that the GMFM-ER should be used as the outcome measure for motor function in children with CP, rather than the GMFM, especially for interventions performed on longer periods, like HBOT. Because the natural evolution of the GMFM-66 score has been previously predicted for children with CP who were less than eight years old [55] and because the precise age of individual children in the different studies included in the present review was not available, we could not compute the GMFM-ER in the meta-analysis. As suggested by Collet et al., an alternative hypothesis to explain significant changes in all groups is that improvements on the GMFM were due to a participation effect (increased interactions at both the children and parents' level) [27]. One interesting result in the study by Lacey et al. [29] is in line with this hypothesis: while no change in cognitive and motor impairments was observed in any group, significant improvements on the PEDI were reported, without between-group differences. The improvements on the PEDI, though significant, remained below 5%, which is less than the considered threshold for clinically meaningful change (considered as 10–11% by the team who conceived and developed the scale [56]).

## Safety

We chose to investigate the adverse events of hyperbaric interventions in the children population overall, rather than in children with CP only. This can be justified by the fact that the known adverse effects of HBOT can be classified as linked to the changes in pressure/volume, to the toxicity of $O_2$, and to the material used [42]. They are therefore not specific to the population of children with CP. Investigating them in the whole paediatric population was chosen in order to increase the number of studies for this part of the search and to provide a better overview of the risks associated with HBOT. Adverse events was higher in children receiving HBOT than in those receiving HBT with 1.3 ATA, with a relative risk of 1.5 in the only study which investigated adverse events systematically and as the primary outcome measure [41]. However, the adverse events were similar in both groups in another RCT [29] and the meta-analysis showed no difference between HBOT and HBA. Middle ear barotrauma was by far the most frequent side effect, and could affect up to 50% of children [41] during HBOT. This is not surprising, as higher pressures were used in the HBOT interventions (1.5 to 1.75 ATA) than in the controls (1.3 ATA). Though not severe, middle ear barotrauma must be considered as a major issue in hyperbaric interventions given its high occurrence.

## Limitations

The present systematic review highlights that, despite more than 30 years of experience, only few studies with good methodological quality have been published that investigated the benefits and adverse effects of HBOT in children with cerebral palsy. The quality of the evidence can be considered as moderate to high though, as the results of the three high-quality RCTs are rather consistent. Besides, the heterogeneity in the control interventions and in the outcomes impeded the conduction of meta-analyses, making the interpretation of the results difficult.

Three of the studies were carried out by the same research team [27, 28, 32] including two RCTs. This can introduce a bias, as the data from the same sample of children have been studied several times, and limits the external validity of the results.

Regarding adverse events, apart from the studies by Muller-Bola et al. [41] and Lacey et al. [29], no published work regarding HBOT and HBA in children with CP assessed adverse events in a systematic manner. No precise information about the type, frequency, and severity of these adverse events was available in these studies. Therefore, the conclusion of conflicting evidence about the increased risk of middle-ear barotrauma in HBOT in children is based on only two studies.

## Conclusion

From this systematic review, there is high-level evidence that HBOT does not improve motor function, cognition, and functional performance in children with CP. The use of HBOT in children with CP is therefore not recommended. The safety of HBOT appears to be fair in this population, as only very few severe adverse effects have been reported in the literature. Middle ear barotrauma of minor severity is a frequent adverse event of HBOT at currently used pressure levels (1.5 to 1.75 ATA).

Although the conclusions from this systematic review appear straightforward, attention should be drawn on the fact that they are dependent on the results of the PICO (Patients, Interventions, Comparisons, Outcomes) framework used. The originality of the present review lies in the critical appraisal of the included studies. The discussion above has stressed various, important aspects of the population, treatment and control interventions, and outcome measures, that can be questioned and examined further, based on the presumed physiological impact of HBOT. Therefore, we acknowledge that some questions raised by the 15-year controversy about HBOT in CP remain unsettled, and that there is a room for improvement in the methodology of experimental studies into hyperbaric interventions in children with cerebral palsy. However, in the absence of evidence to be sought from animal studies (that have been extensively performed) or from adult research (which has also been comprehensive), this systematic review closes the debate and claims against further clinical research into HBOT in children with CP.

## Supporting information

**S1 Checklist. PRISMA checklist.** PRISMA-Checklist including the information associated with the systematic review presented in the manuscript and tables.
(PDF)

**S1 File. PEDRO HBOT.** PEDRO scores of the different RCTs included in the systematic review.
(XLSX)

**S2 File. MDB HBOT.** Modified Downs and Black scores of the non RCT studies included in the systematic review.
(XLSX)

## Author Contributions

**Conceptualization:** Justine Laureau, Raphaël Gross.

**Data curation:** Justine Laureau, Raphaël Gross.

**Investigation:** Justine Laureau, Raphaël Gross.

**Methodology:** Justine Laureau, Christelle Pons, Raphaël Gross.

**Validation:** Christelle Pons, Guy Letellier.

**Writing – original draft:** Justine Laureau, Guy Letellier, Raphaël Gross.

**Writing – review & editing:** Christelle Pons, Guy Letellier, Raphaël Gross.

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
