## [Decision Letter · Decision Letter 0]

18 Dec 2021

PONE-D-21-20366Hyperbaric Oxygen in Children with Cerebral Palsy: A Systematic Review of Effectiveness and Safety.PLOS ONE

Dear Dr. Gross,

Thank you for submitting your manuscript to PLOS ONE. After careful consideration, we feel that it has merit but does not fully meet PLOS ONE’s publication criteria as it currently stands. Therefore, we invite you to submit a revised version of the manuscript that addresses the points raised during the review process.

Both reviewers aknowledged the relevance of the review, however, a recently published meta-analysis was not included since it appeared this last September. It would be appropriate to revise te paper by including also that important work. More minor changes are also requested.

We look forward to receiving your revised manuscript.

Kind regards,

Andrea Martinuzzi

Academic Editor

PLOS ONE

Journal Requirements:

3. PLOS requires an ORCID iD for the corresponding author in Editorial Manager on papers submitted after December 6th, 2016. Please ensure that you have an ORCID iD and that it is validated in Editorial Manager. To do this, go to ‘Update my Information’ (in the upper left-hand corner of the main menu), and click on the Fetch/Validate link next to the ORCID field. This will take you to the ORCID site and allow you to create a new iD or authenticate a pre-existing iD in Editorial Manager. Please see the following video for instructions on linking an ORCID iD to your Editorial Manager account: https://www.youtube.com/watch?v=_xcclfuvtxQ.

Reviewers' comments:

Reviewer's Responses to Questions

**Comments to the Author**

1. Is the manuscript technically sound, and do the data support the conclusions?

Reviewer #1: Partly

Reviewer #2: Yes

2. Has the statistical analysis been performed appropriately and rigorously? 

Reviewer #1: Yes

Reviewer #2: Yes

3. Have the authors made all data underlying the findings in their manuscript fully available?

Reviewer #1: No

Reviewer #2: Yes

4. Is the manuscript presented in an intelligible fashion and written in standard English?

Reviewer #1: Yes

Reviewer #2: Yes

5. Review Comments to the Author

Reviewer #1: This paper aimed to update the scientific information about the impact of hyperbaric oxygen therapy on clinical symptoms of children with cerebral palsy.

The paper is well written, and the authors have an accurate interpretation of the studies included in this review. The search strategy is appropriated, and all papers have been screening correctly. My major concern about this review is the fact that they did not include the meta-analysis, entitled “Hyperbaric Oxygen Therapy Is Beneficial for the Improvement of Clinical Symptoms of Cerebral Palsy: A Systematic Review and Meta-Analysis “, which has just been published in September in complementary research journal. The authors have included article until May 2021, nevertheless they should extend the search period and include this key meta analysis. Moreover, the latter did not reach the same conclusions about the effect of HBOT. In regard to this recent meta-analysis, authors of the present review must argue why is it still relevant to summarize the evidence on this intervention in this population. Also, of course, they had to explain review discrepancies throughout.

Introduction

“Adjuvant” is not appropriated.

“Postulated” is not appropriated.

Which phenomena can be put forward to explain that more oxygen could "switch on" neuronal cells.

Could the authors be more precise about the “enzymatic cascades”

Discussion

Interventions and controls

Authors have to explain which design should be apply in future studies to differentiate effects of hyperpressurization and hyperoxygenation.

Patient

Authors should explain why some subtypes of children with CP (and which subtype) could have a positive response to HBOT?

Development trajectories

Authors should compare the changes in both groups in regard to the GMFM-ER. Is the increase reported in both groups is more than the natural progression?

As proposed by the authors, they should reinterpret the result of the studies based on the GMFM-ER. In that way they could brings relevant information in regards to what have been done before.

Reviewer #2: In this manuscript, the researchers describe the effectiveness of HBOT on children with CP. They achieved this by a meta analysis of the literature.

Overall this is a nice manuscript that integrates nicely several studies on HBOT in CP. It is an important manuscript, that has a significant and important conclusion regarding the lack of effectiveness of HBOT in CP.

The writing is good, and the presentation of data is great.

I only have few comments that will potentially improve the manuscript:

1) In the introduction, more information on CP should be given to the reader, especially by citing more relevant and updated review articles on CP.

2) Similarly, more information on HBOT and how it affects neurological conditions should be cited, shedding light on the biological, molecular, cellular mechanisms that HBOT is acting through. Recent reviews on that matter should be added, to expand the reader's knowledge. Also, HBOT effectiveness in CP, stroke and other neurological conditions should be added, to expand the reader's knowledge on HBOT effectiveness in general.

3) Can the authors further discuss how exactly HBOT helps to motor dysfunction in the introduction?

4) In the results section, it can be helpful if the authors add more description on the effect of HBOT on some aspects. For example, when referring to sleep disorders, "improvement in the quality of sleep" is not informative enough. Can the authors add few words on what parameters and how they were measured? This is relevant to all results section (sleep was just an example).

5) While the conclusions in the manuscript are valid, I suggest the authors emphasize their suggestions and recommendations as to how to improve the future research on HBOT in CP. Strictly define the recommendations regarding the pressure, control groups, range of ages etc. This will increase the impact of the paper and improve the research/treatment field.

6. PLOS authors have the option to publish the peer review history of their article (what does this mean?). If published, this will include your full peer review and any attached files.

Reviewer #1: No

Reviewer #2: **Yes: **Boaz Barak

---

## [Author Response · Author response to Decision Letter 0]

9 Feb 2022

We thank the reviewers for their detailed assessment of our work. 

Please find below (in red) our detailed answers to the comments made by the reviewers on our manuscript “Hyperbaric Oxygen in Children with Cerebral Palsy: A Systematic Review of Effectiveness and Safety.” We thank the reviewers for their detailed assessment of our work. 

Comments to the Author

1. Is the manuscript technically sound, and do the data support the conclusions?

Reviewer #1: Partly

Reviewer #2: Yes

2. Has the statistical analysis been performed appropriately and rigorously?

Reviewer #1: Yes

Reviewer #2: Yes

3. Have the authors made all data underlying the findings in their manuscript fully available?

Reviewer #1: No 

We added to supplementary files in order to provide the best description of the studies included in this systematic review. The two files provide the detailed scoring of the quality of each included article, using the modified Downs and Black score for observational studies, and the Pedro score for randomized controlled trials. Besides, Tables 1 and 2 provide all available information regarding the data from the different studies included in this systematic review, about effectiveness and safety of HBOT. We acknowledge that in many of the included studies, the statistics are poorly described, and we reported them according to the full text and tables of the manuscripts available through databases. Therefore, we believe that all data underlying the findings in the present systematic review have been made available in this revised version of our work.

Reviewer #2: Yes

4. Is the manuscript presented in an intelligible fashion and written in standard English?

Reviewer #1: Yes

Reviewer #2: Yes

5. Review Comments to the Author

Reviewer #1: This paper aimed to update the scientific information about the impact of hyperbaric oxygen therapy on clinical symptoms of children with cerebral palsy.

The paper is well written, and the authors have an accurate interpretation of the studies included in this review. The search strategy is appropriated, and all papers have been screening correctly. My major concern about this review is the fact that they did not include the meta-analysis, entitled “Hyperbaric Oxygen Therapy Is Beneficial for the Improvement of Clinical Symptoms of Cerebral Palsy: A Systematic Review and Meta-Analysis “, which has just been published in September in complementary research journal. The authors have included article until May 2021, nevertheless they should extend the search period and include this key meta analysis. Moreover, the latter did not reach the same conclusions about the effect of HBOT. In regard to this recent meta-analysis, authors of the present review must argue why is it still relevant to summarize the evidence on this intervention in this population. Also, of course, they had to explain review discrepancies throughout.

We thank the reviewer for his/her vigilance, which allowed the identification of this recent systematic review with meta-analysis. We have considered this article with great attention. 

First, we would like to point that it was published in September 2021, while our systematic review was submitted earlier, in July 2021, which explains why the later review by Zhang et al. was not included in our work. It seems that some issues in the edition process were responsible for the delay between our submission in July 2021 and the access to the manuscript for the reviewers.

Second, and more important, we tried to take the results of the studies included in the systematic review by Zhang into account, as suggested. In the Results section of their review and in its reference list, we identified 24 articles that had not been identified by our search strategy (descriptive series of cases, observational studies, and controlled studies into therapeutic effects of hyperbaric oxygen therapy in children with CP: REFERENCES 2, 15-28, and 30-38). We systematically investigated all these 24 references. Unfortunately, it turned out that among those 24, four articles have their full-text available in Chinese only, preventing us from including their results in our systematic review. Indeed, only articles in English were included in our work. Moreover, the remaining 20 references were not available in the usual databases (Scholar, PEDRO, PubMed, Cochrane), making these 20 articles inaccessible. It is beyond the scope of the answers to the reviewers to make hypotheses on the scientific work that these 20 references represent. However, we are sorry to state that we cannot include these original articles in our systematic review. 

Anyone can check that the articles that were included in our systematic review are referenced and available on scientific databases, which is not the case for many of the articles retained in the review by Zhang and published in September 2021. Therefore, we claim that our literature review is comprehensive, and assume that our work has to be considered independently of the review by Zhang, whose conclusion cannot be included in the present work for the aforementioned reasons.

Introduction

“Adjuvant” is not appropriated. 

The sentence “Considered as an adjuvant alternative therapy, …” has been replaced by “ Considered as a complementary or alternative therapy, …

“Postulated” is not appropriated.

The term “postulated” has been replaced by “supposed”

Which phenomena can be put forward to explain that more oxygen could "switch on" neuronal cells.

The following sentence has been added: “HBOT, by increasing dramatically the amount of free oxygen in the blood, could increase the delivery of oxygen to such areas and reveal the activity of quiet neuronal cells.” With reference to [14] McDonagh et al.

Could the authors be more precise about the “enzymatic cascades”.

As it is difficult to describe the whole biochemical processes involved in hypoxic inflammation we suggest to add a summary sentence : “Besides, a chemical enzymatic cascade would be triggered by cellular hypoxia, with a key-role of hypoxia-inducible factor-1a in inflammatory sites with low oxygen levels [19].” And to refer to the review article by Carl Nathan in Nature, 2003 (REF 19).

The end of the Introduction has been modified in order to give a better understanding of the issues about the biological and therapeutic effects of HBOT which remain to date, and justify the present systematic review: “In contrast with these experimental findings in vitro, there is a lack of studies demonstrating the reversibility of the damages induced by anoxia or ischaemia through HBOT in animal models of CP. In children with CP, the evidence regarding a therapeutic effect of HBOT on motor and/or cognitive impairments remains scarce. To the best of our knowledge,, only one literature review has been published (2007) about its effectiveness on motor and cognitive functions in this population [8]. The evidence was considered as inadequate for establishing a significant benefit of HBOT on functional outcomes [8]. The methods as well as the interpretation of the results of studies that were included in this review have raised important controversies, and calls for more studies, addressing the issue of the control treatment, have been made [20]. Since then, the debate remains open and scientists continue to argue about HBOT in those with CP, mostly through editorials and position papers [5–7,20]. Recently, Novak et al. endorsed a firm position against the use of HBOT in children with CP. As several clinical studies have been published since the first review [8], it appears relevant to provide an update of the literature regarding the effects of HBOT in children with CP, in order to collect and summarize the evidence on this intervention in this population.”

Discussion

Interventions and controls

Authors have to explain which design should be apply in future studies to differentiate effects of hyperpressurization and hyperoxygenation.

The following information about a potential therapeutic trial has been added: 

“Theoretically, it is possible to design future studies with an experimental design where the effects of hyperpressurization and hyperoxygenation would be differentiated, by dividing the experimental samples into, for instance, four experimental groups: one group with usual HBOT (1.5 ATM, 100% O2), one group with pure oxygen at normal pressure (1.0 ATM, 100% O2), one group with pressurized air and an equivalent atmospheric oxygen content (1.5 ATM, 14% O2), and one control group with normal air (1.0 ATM, 21% O2).”

Patient

Authors should explain why some subtypes of children with CP (and which subtype) could have a positive response to HBOT?

We added the following part to the sentence: “(It could be proposed in future studies to assess brain perfusion imaging and functional MRI, together with clinical measures, in order to determine if a sub-type of cerebral lesion (leukomalacia, stroke…) could be associated with a positive response to HBOT) … which would be in line with its presumed effect of increasing the delivery of oxygen to the brain .

Besides, the pathophysiological rationale for a potential specific effect of HBOT in some subtypes of CP has been explained in the first part of the paragraph. 

Development trajectories

Authors should compare the changes in both groups in regard to the GMFM-ER. Is the increase reported in both groups is more than the natural progression?

As proposed by the authors, they should reinterpret the result of the studies based on the GMFM-ER. In that way they could brings relevant information in regards to what have been done before.

We thank the reviewer for this comment, which meets our initial attempt to compute the GMFM-ER in order to compare the progression in the GMFM reported in the different studies which used it as an outcome measure, and the spontaneous evolution of these children according to the GMFM-ER. Unfortunately, we stated that this was not possible for two reasons. First, in the study by Marois et al. which established the GMFM-ER, the expected natural evolution of the GMFM-66 score was predicted for any group of children with CP who were under eight years old. Second, the computation of the GMFM-ER for a given child, based on the progression of the GMFM compared to the natural progression, is dependent on the child’s age at baseline. As all subjects in the studies included in our systematic review were not under 8, and as the precise age at inclusion was not available for all subjects in some studies, we could not calculate the GMFM-ER. 

We suggest to add the following sentence in order to explain why the use of the GMFM-ER in the discussion remains a suggestion for future studies, and why we did not compute GMFM-ER values from the available data:

“Because the natural evolution of the GMFM-66 score has been previously predicted for children with CP who were less than eight years old [55] and because the precise age of individual children in the different studies included in the present review was not available, we could not compute the GMFM-ER in the meta-analysis.”

Reviewer #2: In this manuscript, the researchers describe the effectiveness of HBOT on children with CP. They achieved this by a meta analysis of the literature.

Overall this is a nice manuscript that integrates nicely several studies on HBOT in CP. It is an important manuscript, that has a significant and important conclusion regarding the lack of effectiveness of HBOT in CP.

The writing is good, and the presentation of data is great.

I only have few comments that will potentially improve the manuscript:

1) In the introduction, more information on CP should be given to the reader, especially by citing more relevant and updated review articles on CP.

We acknowledge that there is much to say about the recent and ongoing advances in the description, pathophysiology, risk factors, classification, prevention means, and early interventions in children with CP. In order to restrict the number of words in the Introduction section, we suggest to add one reference which is in our opinion a good summary of these issues(Korzeniewski et al. Nature Reviews Neurology 2018). The following sentence has been added: 

“Cerebral palsy has proven to have complex aetiology with many risk factors (only recently and partly unravelled). The definition of subtypes of CP, the potential interventions to prevent the extent of lesions and impairments, and the impact of early interventions is still work in progress [2,3].” 

2) Similarly, more information on HBOT and how it affects neurological conditions should be cited, shedding light on the biological, molecular, cellular mechanisms that HBOT is acting through. Recent reviews on that matter should be added, to expand the reader's knowledge. Also, HBOT effectiveness in CP, stroke and other neurological conditions should be added, to expand the reader's knowledge on HBOT effectiveness in general.

According to the reviewer’s suggestion, the second part of the introduction has been rewritten and now reads as follows:

“In CP, an effect of HBOT has been supposed, which relies on a potential physiological effect of increased oxygen rate and/or pressure on the brain cells around the site of injury. This effect is based on several hypotheses, mostly originating from in vitro studies, animal models of neurological anoxo-ischaemic damage, and human studies into stroke. A first hypothesis relies on the presence of quiet neuronal cells in a so-called “ischaemic penumbra” zone, a volume of tissue surrounding the zone of infarction, where cells are viable and can potentially be “switched on” by appropriate stimuli, among which high doses of oxygen [11,12]. HBOT, by increasing dramatically the amount of free oxygen in the blood, could increase the delivery of oxygen to such areas and reveal the activity of quiet neuronal cells. [13] A second hypothesis is the synaptic sprouting and reorganisation stimulated by the administration of hyperbaric oxygen [13,14]. A third hypothesis is the presence of stem cells within the brain, which would differentiate into functional neurons in the zones of cerebral damage, a process that could be accelerated by the administration of HBOT after an ischaemic insult to the brain [15,16]. Other mechanisms, relying on the ischaemia-induced events, such as angiogenesis and enzymatic cascades, have also been proposed to suggest potential effects of HBOT on pathological conditions at a chronic stage, and therefore on CP [17]. Hypoxia would enhance angiogenesis, which would be potentiated by the supply of additional oxygen [18]. Besides, a chemical enzymatic cascade would be triggered by cellular hypoxia, with a key-role of hypoxia-inducible factor-1a in inflammatory sites with low oxygen levels [19]. Such inflammatory processes induced by anoxia / ischaemia would be down-regulated by HBOT [17]. In contrast with these experimental findings in vitro, there is a lack of studies demonstrating the reversibility of the damages induced by anoxia or ischaemia through HBOT in animal models of CP. In children with CP, the evidence regarding a therapeutic effect of HBOT on motor and/or cognitive impairments remains scarce.” 

3) Can the authors further discuss how exactly HBOT helps to motor dysfunction in the introduction?

According to the changes made in the Introduction and presented in the previous answer to the comment (see above), we think that the effect of HBOT on motor dysfunction in CP and other neurological impairments is now thoroughly described in the Introduction section.

4) In the results section, it can be helpful if the authors add more description on the effect of HBOT on some aspects. For example, when referring to sleep disorders, "improvement in the quality of sleep" is not informative enough. Can the authors add few words on what parameters and how they were measured? This is relevant to all results section (sleep was just an example).

We added specific information about the outcome measures that have been used in the different studies included in the present review:

- For motor function: “Four RCTs reported the effectiveness of HBOT on motor function. All used the GMFM score, considered as the gold-standard for the assessment of motor function in children with cerebral palsy [39].”

- For cognitive functions and language: “Cognitive functions were assessed in three RCTs using various outcome measures [27–29]: Visiospatial and verbal working memory and attention [27,28], or Test of Variables of Attention [29]”;

- For sleep disorders: “The main outcome measure was the total sleep items (TSI) from the Sleep Disturbance Scale for Children Questionnaire [40] in 2 out of 3 studies [36,37]. Both reported a significant reduction of the TSI after HBOT. Montgomery et al. [32] used a custom questionnaire completed by parents, and reported an improvement in the quality of sleep.”

- For spasticity: “One RCT [30] reported a significant reduction of spasticity as assessed by the H/M ratio in the soleus muscle in both the experimental and control groups, without any between-group difference. Two before/after studies [32,38] (m-DB 3-10) reported a reduction of spasticity, assessed clinically through the MAS [32] or a subjective scale [38]”;

- For Independence: “In the two high-quality RCTs [27,29] investigating the functional performance of children with CP using the PEDI (Pediatric Evaluation of Disability Inventory) score,…”. 

- 

5) While the conclusions in the manuscript are valid, I suggest the authors emphasize their suggestions and recommendations as to how to improve the future research on HBOT in CP. Strictly define the recommendations regarding the pressure, control groups, range of ages etc. This will increase the impact of the paper and improve the research/treatment field.

We thank the reviewer for this suggestion. Our recommendations about the potential design of future studies into HBOT in CP have been added to the Discussion section (paragraph Interventions and Controls: 

“it is possible to design future studies with an experimental design where the effects of hyperpressurization and hyperoxygenation would be differentiated, by dividing the experimental samples into, for instance, four experimental groups: one group with usual HBOT (1.5 ATM, 100% O2), one group with pure oxygen at normal pressure (1.0 ATM, 100% O2), one group with pressurized air and an equivalent atmospheric oxygen content (1.5 ATM, 14% O2), and one control group with normal air (1.0 ATM, 21% O2).”

---

## [Decision Letter · Decision Letter 1]

2 May 2022

PONE-D-21-20366R1Hyperbaric Oxygen in Children with Cerebral Palsy: A Systematic Review of Effectiveness and Safety.PLOS ONE

Dear Dr. Gross,

Thank you for submitting your manuscript to PLOS ONE. After careful consideration, we feel that it has merit but does not fully meet PLOS ONE’s publication criteria as it currently stands. Therefore, we invite you to submit a revised version of the manuscript that addresses the points raised during the review process.

Reviewer 3 requires major clarification in the methods section. Please address his concerns detailed in the attached file

We look forward to receiving your revised manuscript.

Kind regards,

Andrea Martinuzzi

Academic Editor

PLOS ONE

Reviewers' comments:

Reviewer's Responses to Questions

**Comments to the Author**

1. If the authors have adequately addressed your comments raised in a previous round of review and you feel that this manuscript is now acceptable for publication, you may indicate that here to bypass the “Comments to the Author” section, enter your conflict of interest statement in the “Confidential to Editor” section, and submit your "Accept" recommendation.

Reviewer #2: All comments have been addressed

Reviewer #3: (No Response)

2. Is the manuscript technically sound, and do the data support the conclusions?

Reviewer #2: Yes

Reviewer #3: (No Response)

3. Has the statistical analysis been performed appropriately and rigorously? 

Reviewer #2: N/A

Reviewer #3: (No Response)

4. Have the authors made all data underlying the findings in their manuscript fully available?

Reviewer #2: Yes

Reviewer #3: (No Response)

5. Is the manuscript presented in an intelligible fashion and written in standard English?

Reviewer #2: Yes

Reviewer #3: (No Response)

6. Review Comments to the Author

Reviewer #2: (No Response)

Reviewer #3: (No Response)

7. PLOS authors have the option to publish the peer review history of their article (what does this mean?). If published, this will include your full peer review and any attached files.

Reviewer #2: **Yes: **Dr. Boaz Barak, Tel Aviv University, Israel

Reviewer #3: **Yes: **Hsiu-Ching Chiu

---

## [Author Response · Author response to Decision Letter 1]

25 May 2022

General comments 

As there are a number of omissions in the design of this study, the value of this topic under review is questionable. 

We have strictly followed the PRISMA guidelines for systematic reviews and meta-analyses and the bodies of evidence were then appraised using the GRADE evidence rating system, which ensures that the methods and results of the present review meet the stat-of-the-art expectations.

Abstract/ Introduction-content

No read.

We assume that there has been no detectable issue in this section.

Methods: Identification and selection of studies

Information is messy. Inclusion criteria should be following as Design(missing), Participants, Intervention (missing), Outcome measures (missing)

We followed the reviewer’s demand and altered this paragraph of the Methods section, which now reads: 

“The following inclusion criteria were used for study selection: 

1) Design: observational studies, before/after studies (descriptive) and controlled trials 

2) Patients: children (<18 years) with CP included in the study sample; 

3) Intervention: administration of HBOT or HBA; 

4) Outcomes: inclusion of an effectiveness- and/or safety-related outcome for HBOT or HBA;

5) Language: articles written in English.”

The title of the paragraph has been changed into the following: Article selection: Identification and selection of studies.

Methods: Assessment of characteristics of studies 

There should be more points in this part. It would be better to have this headings and follow some subheading as below:

Quality: information included , but a bit messy.

Participants: missing

Intervention: messing

Outcome measures: missing

We have taken this comment of the reviewer into account and added the suggested organization in the Methods section. We added the following subheadings: 

“Participants: information about participants (age, type of cerebral palsy, and functional level) were detailed.

Intervention: data regarding the hyperbaric intervention (content: pure oxygen or air, pressure, number and duration of sessions, total volume of the intervention) and control intervention were carefully searched through and reported.

Outcome measures: all types of outcome measures identified in the retained articles were analysed and could subsequently be sorted as: motor function, spasticity, sleep disorders, cognitive functions, independence, for effectiveness-related outcome, and safety.”

Besides, the title of this paragraph of the Methods section has been changed and now reads: “Quality assessment and study appraisal : Assessment of the characteristics of included studies in order to provide more information about the methods used.”

Results/ Discussion/ Conclusion: context.

As there are so many questions in the method, results, discussion and conclusion are questionable.

We are very confident that all informations about the methods that were missing in the reviewer’s opinion have been detailed in the Results section and the associated Tables

Readability and style.

This paper is not easy to read, because the fluency and structure of written English, suggesting for proofreading. 

We asked for a careful assessment of the English in this manuscript by a native English speaker

---

## [Decision Letter · Decision Letter 2]

22 Sep 2022

PONE-D-21-20366R2Hyperbaric Oxygen in Children with Cerebral Palsy: A Systematic Review of Effectiveness and Safety.PLOS ONE

Dear Dr. Gross,

Thank you for submitting your manuscript to PLOS ONE. After careful consideration, we feel that it has merit but does not fully meet PLOS ONE’s publication criteria as it currently stands. Therefore, we invite you to submit a revised version of the manuscript that addresses the points raised during the review process.

In spite of the negative comments of one reviewer (that nevertheless deserve consideration in your next revision), I decided to indicate only minor revision. Please follloe carefully the indications of the reviewer in finalizinng your submission.

We look forward to receiving your revised manuscript.

Kind regards,

Andrea Martinuzzi

Academic Editor

PLOS ONE

Journal Requirements:

Reviewers' comments:

Reviewer's Responses to Questions

**Comments to the Author**

1. If the authors have adequately addressed your comments raised in a previous round of review and you feel that this manuscript is now acceptable for publication, you may indicate that here to bypass the “Comments to the Author” section, enter your conflict of interest statement in the “Confidential to Editor” section, and submit your "Accept" recommendation.

Reviewer #3: (No Response)

Reviewer #4: (No Response)

2. Is the manuscript technically sound, and do the data support the conclusions?

Reviewer #3: No

Reviewer #4: Yes

3. Has the statistical analysis been performed appropriately and rigorously? 

Reviewer #3: No

Reviewer #4: Yes

4. Have the authors made all data underlying the findings in their manuscript fully available?

Reviewer #3: No

Reviewer #4: Yes

5. Is the manuscript presented in an intelligible fashion and written in standard English?

Reviewer #3: No

Reviewer #4: Yes

6. Review Comments to the Author

Reviewer #3: Authors didn't change methods as suggestions earlier.

General comments

As there are a number of omissions in the design of this study, the value of this topic under review is questionable. Authors still didn’t change methods as suggested. It would be difficult to read futher.

Abstract/ Introduction-content

No read

Methods: Identification and selection of studies

Information is messy. Inclusion criteria should be following as Design(missing), Participants, Intervention (missing), Outcome measures (missing)

Methods: Assessment of characteristics of studies

There should be more points in this part. It would be better to have this headings and follow some subheading as below:

Quality: information included , but a bit messy.

Participants: missing

Intervention: messing

Outcome measures: missing

Results/ Discussion/ Conclusion: context.

As there are so many questions in the method, results, discussion and conclusion are questionable.

Readability and style.

This paper is not easy to read, because the fluency and structure of written English, suggesting for proofreading.

Reviewer #4: The comments from a previous review round have overall been appropriately addressed.

The English has been improved, however multiple mistakes or style issues remain, e.g. :

- The definition of subtypes of CP, the potential interventions to prevent the extent of lesions and impairments, and the

impact of early interventions is still work in progress. SHOULD BE CORRECTED TO ...are still work in progress.

- Considered as an complementary  considered as A complementary

- with various oxygen content and pressure  various oxygen contents and pressures OR variable oxygen content and pressure

- etc

It is not the role of the reviewers to provide copy-editing and since a re-read by an English speaker visibly was insufficient I can only advise a round of professional copy-editing before possible publication.

My only comment on content is around the conclusion, which in my opinion unjustifiably keeps a door open to further research on HBOT, which is not supported by this review in any way. The authors state : Therefore, we acknowledge that some questions raised by the 15-year controversy about HBOT in CP remain unsettled, and that there is a room for improvement in the methodology of experimental studies into hyperbaric interventions in children with cerebral palsy. However, encouraging results from animal studies and/or strong evidence in other neurological patients with hypoxic/anoxic brain lesions appear necessary before clinical research in children with CP can be revitalised.

I would contend that there is no additional evidence to be sought from animal studies (that have been extensively performed) or from adult research (which has also been comprehensive) and that therefore this systematic review closes the debate.

7. PLOS authors have the option to publish the peer review history of their article (what does this mean?). If published, this will include your full peer review and any attached files.

Reviewer #3: **Yes: **Hsiu-Ching Chiu

Reviewer #4: No

---

## [Author Response · Author response to Decision Letter 2]

28 Sep 2022

REVIEWER #3

General comments 

As there are a number of omissions in the design of this study, the value of this topic under review is questionable. Authors still didn’t change methods as suggested. It would be difficult to read futher.

We have strictly followed the PRISMA guidelines for systematic reviews and meta-analyses and the bodies of evidence were then appraised using the GRADE evidence rating system, which ensures that the methods and results of the present review meet the stat-of-the-art expectations.

Abstract/ Introduction-content

No read.

We assume that there has been no detectable issue in this section.

Methods: Identification and selection of studies

Information is messy. Inclusion criteria should be following as Design(missing), Participants, Intervention (missing), Outcome measures (missing)

We followed the reviewer’s demand in our previous revision. The recommendations by the reviewer have been rigorously followed. The paragraph of the Methods section now reads: 

“The following inclusion criteria were used for study selection: 

1) Design: observational studies, before/after studies (descriptive) and controlled trials 

2) Patients: children (<18 years) with CP included in the study sample; 

3) Intervention: administration of HBOT or HBA; 

4) Outcomes: inclusion of an effectiveness- and/or safety-related outcome for HBOT or HBA;

5) Language: articles written in English.”

The title of the paragraph has been changed into the following: Article selection: Identification and selection of studies.

Methods: Assessment of characteristics of studies 

There should be more points in this part. It would be better to have this headings and follow some subheading as below:

Quality: information included , but a bit messy.

Participants: missing

Intervention: messing

Outcome measures: missing

We have already taken this comment of the reviewer into account in our previous revision, and added the suggested organization in the Methods section. We added the following subheadings: 

“Participants: information about participants (age, type of cerebral palsy, and functional level) were detailed.

Intervention: data regarding the hyperbaric intervention (content: pure oxygen or air, pressure, number and duration of sessions, total volume of the intervention) and control intervention were carefully searched through and reported.

Outcome measures: all types of outcome measures identified in the retained articles were analysed and could subsequently be sorted as: motor function, spasticity, sleep disorders, cognitive functions, independence, for effectiveness-related outcome, and safety.”

Besides, the title of this paragraph of the Methods section has been changed and now reads: “Quality assessment and study appraisal: Assessment of the characteristics of included studies in order to provide more information about the methods used.”

Results/ Discussion/ Conclusion: context.

As there are so many questions in the method, results, discussion and conclusion are questionable.

We are very confident that all informations about the methods that were missing in the reviewer’s opinion have been detailed in the Results section and the associated Tables

Readability and style.

This paper is not easy to read, because the fluency and structure of written English, suggesting for proofreading. 

We asked for a careful assessment of the English in this manuscript by a native English speaker. 

REVIEWER #4

Reviewer #4: The comments from a previous review round have overall been appropriately addressed.

The English has been improved, however multiple mistakes or style issues remain, e.g. :

- The definition of subtypes of CP, the potential interventions to prevent the extent of lesions and impairments, and the

impact of early interventions is still work in progress. SHOULD BE CORRECTED TO ...are still work in progress.

- We thank the Reviewer for her/his careful reading. We have changed the verb as suggested. 

- Considered as an complementary  considered as A complementary

- We thank the Reviewer for her/his careful reading. We have changed the formulation as suggested. 

- with various oxygen content and pressure  various oxygen contents and pressures OR variable oxygen content and pressure

- We thank the Reviewer for her/his careful reading. We have changed the adjective as suggested, into “variable”. 

It is not the role of the reviewers to provide copy-editing and since a re-read by an English speaker visibly was insufficient I can only advise a round of professional copy-editing before possible publication.

- An additional reading by a native-english speaker has been performed for this revised version.

My only comment on content is around the conclusion, which in my opinion unjustifiably keeps a door open to further research on HBOT, which is not supported by this review in any way. The authors state : Therefore, we acknowledge that some questions raised by the 15-year controversy about HBOT in CP remain unsettled, and that there is a room for improvement in the methodology of experimental studies into hyperbaric interventions in children with cerebral palsy. However, encouraging results from animal studies and/or strong evidence in other neurological patients with hypoxic/anoxic brain lesions appear necessary before clinical research in children with CP can be revitalised.

I would contend that there is no additional evidence to be sought from animal studies (that have been extensively performed) or from adult research (which has also been comprehensive) and that therefore this systematic review closes the debate.

- we agree with the reviewer on this point. In order to have the conclusion in straight line with the discussion, we altered the final part into: “However, in the absence of evidence to be sought from animal studies (that have been extensively performed) or from adult research (which has also been comprehensive), this systematic review closes the debate and claims against further clinical research into HBOT in children with cerebral palsy”

---

## [Editor Report · Decision Letter 3]

29 Sep 2022

Hyperbaric Oxygen in Children with Cerebral Palsy: A Systematic Review of Effectiveness and Safety.

PONE-D-21-20366R3

Dear Dr. Gross,

We’re pleased to inform you that your manuscript has been judged scientifically suitable for publication and will be formally accepted for publication once it meets all outstanding technical requirements.

Kind regards,

Andrea Martinuzzi

Academic Editor

PLOS ONE
---

## [Editor Report · Acceptance letter]

5 Oct 2022

PONE-D-21-20366R3 

Hyperbaric Oxygen in Children with Cerebral Palsy: A Systematic Review of Effectiveness and Safety. 

Dear Dr. Gross:

I'm pleased to inform you that your manuscript has been deemed suitable for publication in PLOS ONE. Congratulations! Your manuscript is now with our production department. 

Kind regards, 

on behalf of

Dr. Andrea Martinuzzi 

Academic Editor

PLOS ONE